# Comparing representations of biological data learned with different AI paradigms, augmenting and cropping strategies

**Andrei Dmitrenko**[1,2]                                                    DMITRENKO@IMSB.BIOL.ETHZ.CH
**Mauro M. Masiero**[1,2]                                                        MASIERO@IMSB.BIOL.ETHZ.CH
[1] *Life Science Zurich PhD Program on Systems Biology*

**Nicola Zamboni**[2]                                                          ZAMBONI@IMSB.BIOL.ETHZ.CH
[2] *Institute of Molecular Systems Biology, ETH Zürich*

**Editors:** Under Review for MIDL 2022

## Abstract

Recent advances in computer vision and robotics enabled automated large-scale biological image analysis. Various machine learning approaches have been successfully applied to phenotypic profiling. However, it remains unclear how they compare in terms of biological feature extraction. In this study, we propose a simple CNN architecture and implement 4 different representation learning approaches. We train 16 deep learning setups on the 770k cancer cell images dataset under identical conditions, using different augmenting and cropping strategies. We compare the learned representations by evaluating multiple metrics for each of three downstream tasks: i) distance-based similarity analysis of known drugs, ii) classification of drugs versus controls, iii) clustering within cell lines. We also compare training times and memory usage. Among all tested setups, multi-crops and random augmentations generally improved performance across tasks, as expected. Strikingly, self-supervised (implicit contrastive learning) models showed competitive performance being up to 11 times faster to train. Self-supervised regularized learning required the most of memory and computation to deliver arguably the most informative features. We observe that no single combination of augmenting and cropping strategies consistently results in top performance across tasks and recommend prospective research directions.

**Keywords:** Representation learning, self-supervised learning, regularized learning, comparison, memory constraints, cancer research, microscopy imaging.

## 1. Introduction

With recent advances in robotics and deep learning methods, automated large-scale biological image analysis has become possible. Different microscopy technologies allow to collect imaging data of samples under various treatment conditions. Then, images are processed to extract meaningful biological features and compare samples across cohorts. As opposed to carefully engineered features used in the past, deep learning approaches that automatically distil relevant information directly from the data are now widespread (Moen et al., 2019).

A lot of approaches, following different paradigms of machine learning, have been successfully applied to image-based phenotypic profiling: from fully supervised approaches (Godinez et al., 2017; Kraus et al., 2017) to generative adversarial learning (Hu et al., 2019; Goldsborough et al., 2017; Radford et al., 2016) and self-supervision (Robitaille et al., 2021; Zhang et al., 2020). However, it remains unclear how these approaches align with each

other in terms of biological feature extraction. The direct comparison is close to impossible, as many aspects differ between the studies: imaging technologies, datasets, learning approaches and model architectures, implementations and hardware. We discuss related works in more depth in **Appendix A**.

In the emergent field of self-supervised learning, a key role of random data augmentations and multiple image views has recently been shown (Caron et al., 2021). Their synergetic impact on learning image representations has not yet been rigorously studied. In this paper, we compare different deep learning setups in their ability to learn representations of drug-treated cancer cells. We propose a simple CNN architecture and implement several approaches to learn representations: the weakly-supervised learning (WSL), the implicit contrastive learning (ICL) and classical self-supervised learning without (SSL) and with regularization (SSR). We train four models on the same dataset of 770k images of cancer cells growing in 2D cultures in a drug screening campaign. We use four settings for each model: with and without random augmentations, with single and multi-crops. The other training conditions are kept identical. We compare the learned representations in three downstream analysis tasks, discuss their performance and provide the comparison summary table. Therefore, our main contributions are:

- implementations of 16 deep learning setups, including state-of-the-art methods trainable within limited resources (the source code and the trained models are available at <https://github.com/dmitrav/morpho-learner>),

- a systematic comparison of learned representations.

## 2. Data

The initial dataset comprises 1.1M high-resolution grey-scale images of drug-treated cancer cell populations growing adherently *in vitro*. It captures 693 unique combinations of 21 cell lines and 31 drugs at 5 different drug concentrations, multiple time points and biological replicates. Details are given in **Appendix H**.

We carefully subset the initial data to obtain a balanced dataset of two labels: strong drug effect (i.e., the highest drug concentration, the latest time point) and control (no drugs, any time point). We end up with about 770k image crops of size $64 \times 64$. It is important to note that some drugs did not provoke any effect on resistant cell lines, so the corresponding images of drugs and controls look similar. Some other drugs showed growth arrest only, which re-

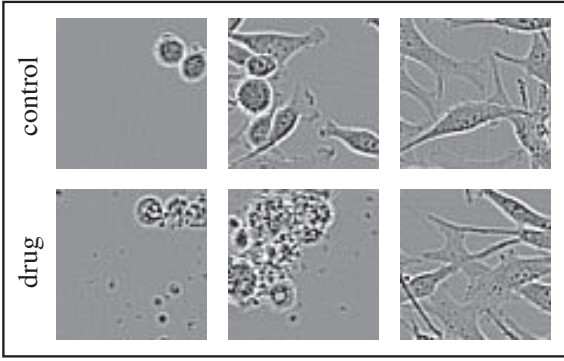

Figure 1: On the left, an early time point of the control (cells have not grown yet) is shown against a strong drug effect (fragmented cells). On the right, the end time points for the control and an ineffective drug are depicted. In the middle, an intermediate growth of the control versus another cytotoxic drug is given.

sulted in drug-treated images being similar to early time point controls, where the cells have

not grown yet. By balancing the dataset to contain such cases (**Figure 1**), we expected the models to learn specific morphological differences, instead of superficial features like cell location on the crop, cell population density, amount of grey, etc.

## 3. Methods

### 3.1. Model architectures

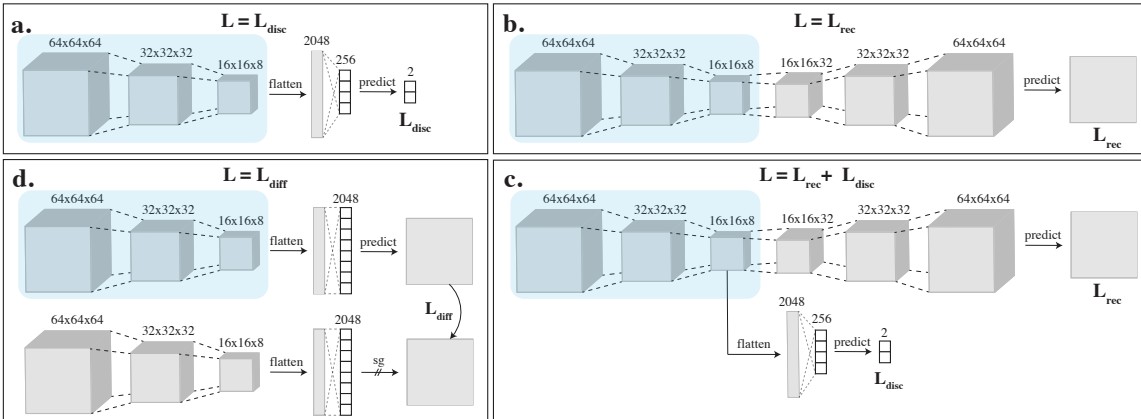

Figure 2: Graphical overview of models. **a.** A weakly-supervised deep classifier with a categorical cross-entropy discrimination loss. **b.** A convolutional autoencoder with a binary cross-entropy reconstruction loss as in Creswell et al. (2017). **c.** A regularized convolutional autoencoder: models a and b, sharing the CNN backbone, trained simultaneously. **d.** A self-supervised CNN backbone with a mean squared error difference loss.

To learn image representations, we implemented the following models:

a. deep classifier with two output labels only (WSL),

b. convolutional autoencoder with classic encoder-decoder architecture (SSL),

c. models **a** and **b** in the joint encoder-classifier-decoder architecture (SSR),

d. CNN backbone, trained with BYOL (Grill et al., 2020) (ICL).

As seen on **Figure 2**, the four architectures contain the same CNN backbone, which is used to produce image representations for the downstream analysis. It was important to use the same stack of layers to ensure fair comparison of methods. However, image representations are learned solving substantially different tasks.

For the SSR model, we adopted a particular implementation where a classifier and an autoencoder are trained in turns, optimizing different loss functions (**Figure 2c**). Our idea was to encourage the autoencoder to learn representations that would bear differences between drug and control images, while still delivering high quality image reconstructions. In this formulation, the classifier acts as a regularizer. Although similar models have been utilized in chemo- and bioinformatics tasks (Gómez-Bombarelli et al., 2018; Rong et al., 2020), to our knowledge this architecture has not been tested previously in the analysis of biological images.

### 3.2. Training setups

Each model was trained under 4 conditions of presence and absence of random image augmentations and multi-crops, giving rise to 16 training setups in total. Since the dataset is naturally grayscale, we only applied random resized crops, horizontal flips and Gaussian blurs to augment. Note that data augmentations are intrinsic to the self-supervised approach. Therefore, we tested single and double augmenting (while preprocessing and/or while training) for the ICL model. In the one-crop setting, we used single $64 \times 64$ images. For the multi-crop setting, we added 4 random resized crops applied to $64 \times 64$ images: 2 of about half-size, and 2 more of about quarter-size (5 crops in total).

We implemented the 16 described setups and trained them using Nvidia GeForce RTX 2060 with 6 GB only. We chose the CNN backbone architecture, batch size and other common hyperparameters by running grid search and finding the best average performance across models, achievable within reasonable training time and hardware memory constraints.

For the ICL model, we additionally optimized BYOL parameters: *projection_size*, *projections_hidden_size* and *moving_average_decay*. We trained the model 100 times, sampling parameters from predefined ranges. We found that equal number of neurons for hidden and projection layers consistently delivered the lowest MSE loss.

We trained all models for 50 epochs, using Adam optimizer with a constant learning rate of 0.0001. A batch size of 256 was used. We defined the same early stopping criterion, which checks a simple divergence condition on the loss function. We used the same data splits with 10% for the validation set to test classification accuracy and reconstruction quality.

### 3.3. Validation and evaluation

We validated the models by monitoring loss functions, classification accuracy and image reconstruction quality for training and validation sets (**Appendix B**). Evaluation metrics for the downstream tasks are described below.

#### 3.3.1. Distance metrics

First, we compared the learned representations in their ability to capture similarity of known drugs. Let $S_1$ and $S_2$ be the sets of images of two drugs known to have similar effects and, $C$ be the set of control images. We quantify similarity between $S_1$ and $S_2$ as follows:

- $D(S_1, S_2) = \underset{u \in S_1, v \in S_2}{\mathrm{median}} (||u - v||)$,
  i.e., the median Euclidean distance between any two images $(u, v)$ of two sets.

- $d(S_1, S_2) = \frac{\hat{D} - D(S_1, S_2)}{\hat{D}}$, where $\hat{D} = \frac{1}{2}[D(S_1, C) + D(S_2, C)]$,
  i.e., the normalized difference between drug-to-control and drug-to-drug distances.

#### 3.3.2. Classification metrics

Next, we performed binary classification. We used a pretrained stack of layers of each model to generate latent representations and then trained a two-layer classifier to differentiate between drugs and controls. We used the same data splits and trained for 25 epochs with SGD optimizer and batch size of 1024. We ran grid search over learning rate, momentum

and weight decay to achieve the best validation accuracy. To comprehensively evaluate the performance, we calculated several metrics individually for each cell line: accuracy, precision, recall and area under ROC.

### 3.3.3. Clustering metrics

Finally, we performed clustering to quantify how similar drug effects group in the latent space. For each cell line, we obtained image representations, reduced their dimensionality with UMAP (McInnes et al., 2020), and clustered their embeddings with HDBSCAN (McInnes et al., 2017). We evaluated several metrics on the partitions: number of identified clusters and percent of noise points, Silhouette score and Davies-Bouldin similarity.

For each cell line, we ran grid search over two parameters: i) *n_neighbors*, responsible for constraining the size of local neighborhood in UMAP, and ii) *min_cluster_size*, representing the smallest grouping size in HDBSCAN. We adopted the following procedure to find the best partitions: 1) select Silhouette scores above median, 2) select Davies-Bouldin scores below median, 3) select the lowest percent of noise, 4) pick a parameter set of max number of clusters. This logic was motivated by zero correlation between Silhouette and Davies-Bouldin measures, and by the objective to find as many "clean" clusters as possible.

## 4. Results

### 4.1. Distance-based drug similarity analysis

Pemetrexed (PTX) and Methotrexate (MTX) are two drugs that have similar chemical structures and both inhibit folate-related enzymes. Over the years, they have been successfully applied to cure many types of cancer (Ruszkowski et al., 2019). We applied distance-based analysis to evaluate how close PTX and MTX are to each other in terms of learned features, and how distant they both are from controls (images of cells under no treatment).

We picked all images related to PTX and MTX drugs from the validation set. Then, we randomly picked the same number of control images (DMSO). We calculated $D(MTX, PTX)$, $D(MTX, DMSO)$, $D(PTX, DMSO)$ on image representations, which resulted in around 3600 distances for each cell line and pair on average. Based on *a-priori* knowledge of efficiency and similarity of the drugs, we expected MTX-PTX distances to be consistently lower than of MTX-DMSO and PTX-DMSO. Analysis of M14 cell line shows it was not the case for all models (**Appendix C**).

We repeated the same analysis for each of 21 cell lines. We found that with the exception of the WSL model, all produced lower average MTX-PTX distances, compared to MTX-DMSO and PTX-DMSO. This suggests that the space of learned features of the WSL model is likely to contain more trivial information about the drug effects, rather than features of altered morphology. Interestingly, the median normalized difference $d$ turned out to be the largest for the ICL model (**Table 2**).

### 4.2. Classification of drugs versus controls

All models showed comparable classification performance, crossing 0.6 accuracy bottom line and reaching 0.7 in many cases. However, it is only the WSL model that achieved 0.8 accuracy for some cell lines (**Appendix F**) and delivered consistently higher performance

in all setups. This was expected due to identical problem formulation during representation learning. Notably, the other three models have shown rival performance on this task. That implies that all models have a potential in detecting drug effects in time-series imaging data (e.g., to predict drug onset times for different concentrations).

**Table 2** contains four classification metrics for each training setup, evaluated on the entire dataset. Median performance on 21 cell lines is reported. The SSR model with single crops and augmentations showed the highest overall accuracy ($0.76 \pm 0.07$) and ROAUC ($0.76 \pm 0.06$), though the WSL model was the most robust across settings. The WSL and ICL models improved performance with multi-crops.

### 4.3. Clustering analysis within cell lines

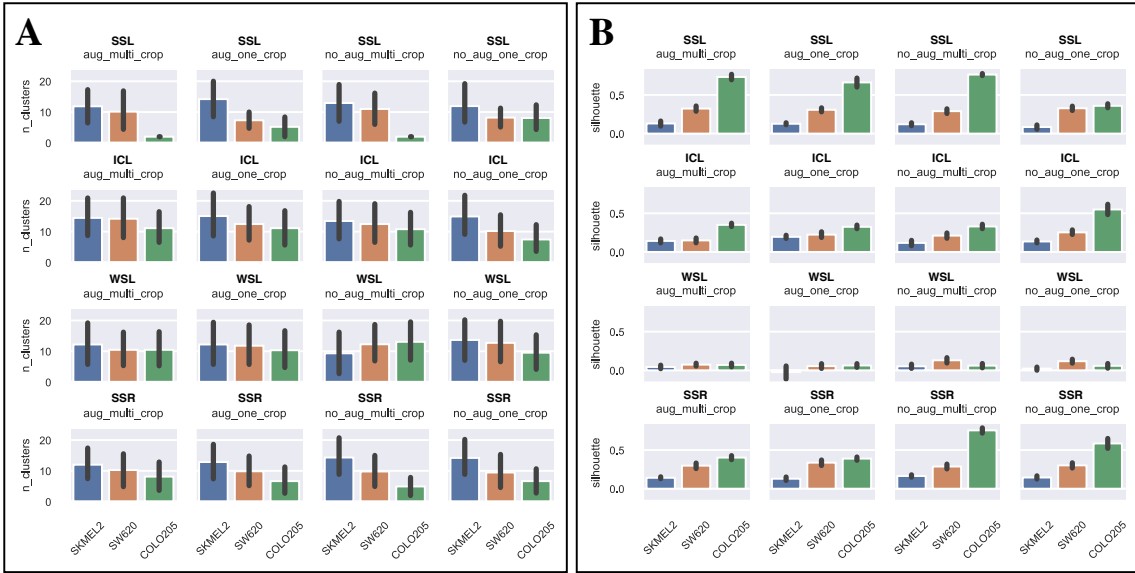

Figure 3: Clustering analysis for three picked cell lines: SKMEL2, SW620, COLO205. Mean numbers of identified clusters (**A**) and mean Silhouette scores (**B**) are shown with confidence intervals. Model architectures are (from top to down): SSL, ICL, WSL and SSR. Different setups are (from left to right): augmentations + multi-crops, augmentations + single crops, no augmentations + multi-crops, no augmentations + single crops.

**Figure 3A** presents numbers of identified clusters across models and settings. Varying the clustering parameters resulted in relatively large confidence intervals. However, even the lower bounds exceeded n=2 clusters, which would correspond to the trivial case of differentiating between drugs and controls (effect vs no effect), in the majority of cases. That indicates that the learned representations allow studying the data in more depth (e.g., finding similarities in concentration-dependent morphological drug effects).

Although mean numbers of clusters look similar, the quality of partitions differed substantially across cell lines, as follows from the Silhouette score barplots (**Figure 3B**). The WSL model produced the poorest scores for the three picked cell lines. Close-to-zero and even negative values suggest that the clusters were mainly overlapping. In such cases, obtained partitions are far less trustworthy and any follow-up analysis on them is controversial.

The top performance was shown by the SSL and the SSR models. Statistics across all cell lines are given in **Table 2**.

### 4.4. Training times and memory usage

All models were trained using Nvidia GeForce RTX 2060 with 6 GB memory. With batch size of 256, steady memory consumption was around 4.3 and 4.7 GB for single and multi-crops, respectively. Batch size of 512 resulted in cuda-out-of-memory error in all setups.

Unlike memory usage, training times differed largely for model architectures and cropping strategies (**Table 1**). The ICL model was the only one to meet the early stopping criterion, which resulted in *remarkably* small training times. The `one_crop`

Table 1: Training time (hours)

|  | SSL | ICL | WSL | SSR |
|---|---|---|---|---|
| one_crop | 7 | **1.5** | 2.5 | 9 |
| multi_crop | 35 | **4** | 11 | 45 |

training stopped after 16/50 epochs, whereas `multi_crop` made 7/50 epochs only. The other models were trained for all 50 epochs.

## 5. Conclusion

We applied different AI paradigms to learn representations of images of drug treated cancer cell lines. We implemented and trained 16 deep learning setups under identical conditions to ensure fair comparison of learned representations. We evaluated them on 3 tasks using multiple metrics to quantify performance. We made the following observations:

- Multi-crops and augmentations generally improve performance in downstream tasks, as expected. Of 40 rows in the comparison **Table 2**, only 6 show superior performance with no augmentations and single crops (bold values in the rightmost column only).

- The CNN backbone trained with BYOL (ICL) showed competitive performance and was the fastest to train. Strikingly, we managed to train it on the 770k dataset using a moderate GPU within 1.5 and 4 hours only (for single and multi-crops). Additionally, double augmenting resulted in improved performance on 2 of 3 downstream tasks.

- Overall, the regularized autoencoder (SSR) produced the most informative features. It delivered the best accuracy and ROAUC in the classification task and the best quality of partitions in the clustering task. However, it required more time to train.

- No single combination of model (architecture) and setting (augmenting and cropping strategy) consistently outperformed the others. Within each model, the top performance on downstream tasks was often shown by different settings.

Our results suggest a combination of contrastive learning and domain-specific regularization as the most promising way to efficiently learn semantically meaningful representations. To achieve top performance in a particular application, we recommend to extensively evaluate the strength of regularization, as well as augmenting and cropping strategies.

Table 2: Summary of comparison. Median drug similarity distances and drug-vs-control classification metrics are given with median absolute deviations. Mean clustering analysis metrics are given with standard deviations. All metrics satisfy *the higher the better*. Top performance for each model and task is highlighted in bold.

| | SSL | | | |
|---|---|---|---|---|
| | aug | | no_aug | |
| | multi_crop | one_crop | multi_crop | one_crop |
| $d$(MTX, PTX) | $0.17 \pm 0.00$ | $0.17 \pm 0.00$ | $0.16 \pm 0.00$ | $\mathbf{0.20 \pm 0.00}$ |
| $D^{-1}$(MTX, PTX) | $\mathbf{0.11 \pm 0.02}$ | $0.08 \pm 0.01$ | $\mathbf{0.11 \pm 0.02}$ | $0.09 \pm 0.01$ |
| Accuracy | $0.72 \pm 0.06$ | $0.70 \pm 0.06$ | $0.72 \pm 0.05$ | $\mathbf{0.75 \pm 0.07}$ |
| Precision | $0.80 \pm 0.06$ | $0.75 \pm 0.05$ | $0.75 \pm 0.04$ | $\mathbf{0.86 \pm 0.06}$ |
| Recall | $0.66 \pm 0.11$ | $0.69 \pm 0.10$ | $\mathbf{0.72 \pm 0.11}$ | $0.65 \pm 0.12$ |
| ROAUC | $0.72 \pm 0.06$ | $0.69 \pm 0.06$ | $0.70 \pm 0.05$ | $\mathbf{0.75 \pm 0.06}$ |
| # clusters | $\mathbf{4 \pm 2}$ | $\mathbf{4 \pm 2}$ | $\mathbf{4 \pm 2}$ | $3 \pm 1$ |
| Not noise, % | $93 \pm 6$ | $93 \pm 5$ | $\mathbf{94 \pm 5}$ | $\mathbf{94 \pm 5}$ |
| Silhouette | $0.32 \pm 0.14$ | $0.34 \pm 0.17$ | $\mathbf{0.35 \pm 0.16}$ | $0.32 \pm 0.08$ |
| (Davies-Bouldin)$^{-1}$ | $0.92 \pm 0.79$ | $\mathbf{0.99 \pm 0.83}$ | $0.94 \pm 0.89$ | $0.80 \pm 0.27$ |
| | ICL | | | |
| $d$(MTX, PTX) | $\mathbf{0.27 \pm 0.00}$ | $0.24 \pm 0.00$ | $0.25 \pm 0.00$ | $0.2 \pm 0.00$ |
| $D^{-1}$(MTX, PTX) | $0.22 \pm 0.02$ | $\mathbf{0.69 \pm 0.06}$ | $0.26 \pm 0.02$ | $0.64 \pm 0.09$ |
| Accuracy | $\mathbf{0.62 \pm 0.05}$ | $0.60 \pm 0.04$ | $0.61 \pm 0.04$ | $0.61 \pm 0.05$ |
| Precision | $\mathbf{0.69 \pm 0.05}$ | $0.63 \pm 0.04$ | $\mathbf{0.69 \pm 0.05}$ | $\mathbf{0.69 \pm 0.05}$ |
| Recall | $0.54 \pm 0.14$ | $\mathbf{0.63 \pm 0.10}$ | $0.56 \pm 0.12$ | $0.55 \pm 0.13$ |
| ROAUC | $\mathbf{0.62 \pm 0.04}$ | $0.59 \pm 0.03$ | $0.61 \pm 0.04$ | $0.61 \pm 0.05$ |
| # clusters | $\mathbf{5 \pm 3}$ | $4 \pm 3$ | $4 \pm 2$ | $3 \pm 1$ |
| Not noise, % | $93 \pm 4$ | $94 \pm 4$ | $\mathbf{95 \pm 5}$ | $\mathbf{95 \pm 4}$ |
| Silhouette | $0.29 \pm 0.09$ | $0.32 \pm 0.06$ | $\mathbf{0.34 \pm 0.09}$ | $\mathbf{0.34 \pm 0.12}$ |
| (Davies-Bouldin)$^{-1}$ | $0.74 \pm 0.15$ | $0.75 \pm 0.14$ | $0.86 \pm 0.47$ | $\mathbf{0.92 \pm 0.63}$ |
| | WSL | | | |
| $d$(MTX, PTX) | $-0.15 \pm 0.00$ | $\mathbf{0.03 \pm 0.00}$ | $-0.18 \pm 0.00$ | $0.01 \pm 0.00$ |
| $D^{-1}$(MTX, PTX) | $0.14 \pm 0.03$ | $\mathbf{1.47 \pm 0.26}$ | $0.1 \pm 0.02$ | $1.20 \pm 0.19$ |
| Accuracy | $0.73 \pm 0.05$ | $0.73 \pm 0.05$ | $\mathbf{0.75 \pm 0.05}$ | $0.73 \pm 0.05$ |
| Precision | $0.73 \pm 0.05$ | $\mathbf{0.75 \pm 0.04}$ | $\mathbf{0.75 \pm 0.04}$ | $\mathbf{0.75 \pm 0.04}$ |
| Recall | $\mathbf{0.77 \pm 0.12}$ | $0.75 \pm 0.11$ | $\mathbf{0.77 \pm 0.11}$ | $\mathbf{0.77 \pm 0.10}$ |
| ROAUC | $0.72 \pm 0.05$ | $0.73 \pm 0.05$ | $\mathbf{0.74 \pm 0.05}$ | $0.73 \pm 0.05$ |
| # clusters | $5 \pm 4$ | $3 \pm 1$ | $4 \pm 1$ | $\mathbf{6 \pm 5}$ |
| Not noise, % | $90 \pm 5$ | $\mathbf{91 \pm 7}$ | $89 \pm 8$ | $87 \pm 7$ |
| Silhouette | $0.13 \pm 0.08$ | $\mathbf{0.14 \pm 0.09}$ | $0.13 \pm 0.08$ | $0.12 \pm 0.07$ |
| (Davies-Bouldin)$^{-1}$ | $0.55 \pm 0.17$ | $0.54 \pm 0.14$ | $\mathbf{0.56 \pm 0.17}$ | $0.53 \pm 0.10$ |
| | SSR | | | |
| $d$(MTX, PTX) | $0.17 \pm 0.00$ | $\mathbf{0.19 \pm 0.00}$ | $0.15 \pm 0.00$ | $0.18 \pm 0.00$ |
| $D^{-1}$(MTX, PTX) | $\mathbf{0.12 \pm 0.02}$ | $0.08 \pm 0.01$ | $0.09 \pm 0.02$ | $0.08 \pm 0.01$ |
| Accuracy | $0.73 \pm 0.07$ | $\mathbf{0.76 \pm 0.07}$ | $0.70 \pm 0.05$ | $0.72 \pm 0.06$ |
| Precision | $0.79 \pm 0.05$ | $\mathbf{0.83 \pm 0.05}$ | $0.75 \pm 0.04$ | $0.80 \pm 0.06$ |
| Recall | $\mathbf{0.70 \pm 0.11}$ | $0.68 \pm 0.11$ | $0.66 \pm 0.11$ | $0.66 \pm 0.09$ |
| ROAUC | $0.73 \pm 0.06$ | $\mathbf{0.76 \pm 0.06}$ | $0.69 \pm 0.05$ | $0.72 \pm 0.06$ |
| # clusters | $\mathbf{4 \pm 1}$ | $\mathbf{4 \pm 2}$ | $3 \pm 1$ | $\mathbf{4 \pm 2}$ |
| Not noise, % | $93 \pm 5$ | $93 \pm 4$ | $\mathbf{94 \pm 5}$ | $\mathbf{94 \pm 5}$ |
| Silhouette | $0.32 \pm 0.06$ | $0.30 \pm 0.09$ | $\mathbf{0.35 \pm 0.15}$ | $0.33 \pm 0.12$ |
| (Davies-Bouldin)$^{-1}$ | $0.76 \pm 0.20$ | $0.77 \pm 0.26$ | $\mathbf{0.99 \pm 0.92}$ | $0.94 \pm 0.68$ |
| **Total bold** | 11 (20%) | 14 (25.5%) | **16 (29%)** | 14 (25.5%) |

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

## Appendix A. Related work

Weak supervision has been a popular choice to learn medical image representations and has proven its efficiency (Caicedo et al., 2018; Lu et al., 2020). When analyzing samples of different experimental conditions, those are often used as weak labels. In our case, there are 693 combinations of drugs and cell lines. However, the respective drug effects are largely unknown, so we restricted ourselves to using two labels only: drug vs control.

A recent approach to understand morphological features of cancer cells by Longden et al. (2021) takes a self-supervised perspective. The authors apply a deep autoencoder to learn 27 continuous morphological features. However, instead of raw images, they use 624 extracted numerical features as input, which inevitably leads to information loss. In this work, we used a CNN model to learn more features directly from the data.

More examples of self-supervision successfully applied to cell segmentation, annotation and clustering tasks are available (Lu et al., 2019; Santos-Pata et al., 2021). Besides, a contrastive learning framework has been rrecently proposed by Ciortan and Defrance (2021) to learn representations of scRNA-seq data. The authors follow the idea of SimCLR (Chen et al., 2020) and show state-of-the-art (SOTA) performance on clustering task. In this work, we trained a CNN backbone following BYOL (Grill et al., 2020). Unlike SimCLR, this approach does not need negative pairs, yet it was shown to have a superior performance.

Finally, several approaches for learning representations of cell images are based on generative adversarial networks (Arjovsky et al., 2017; Gulrajani et al., 2017). Such models often have two components: the generator and the discriminator networks trained simultaneously in a competitive manner. In this work, we implemented a similar idea in the form of regularization.

Despite the great interest in ways to learn representations of biological data, there have been very few attempts to fairly compare those. A comparison of methods predicting cell functions has come out lately (Padi et al., 2020). However, it was primarily focused on collating traditional machine learning and deep learning. Brief general comparisons can be found in reviews and surveys (Moen et al., 2019; Chandrasekaran et al., 2021; Nguyen et al., 2019), but they lack details and cannot inform decision making. Recently, a thorough comparison of data-efficient image classification models has been published by Brigato et al. (2021). Eventually, this work tackles classification tasks only.

## Appendix B. Quality of image reconstructions

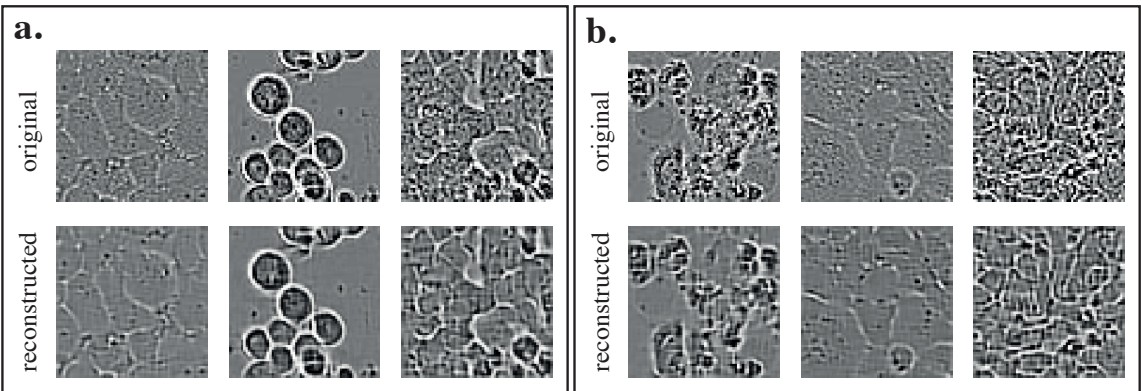

Figure 4: Random examples of reconstructed and original images for the unsupervised (**a**) and the regularized (**b**) models. Regularization did not harm the quality of reconstructions. The learning capacity of the CNN backbone was sufficient to capture normal and altered morphology of the cells.

## Appendix C. Distance-based similarity analysis for M14

Analyzing distances for M14 cell line, we observed that in the latent space the two drugs (PTX and MTX) were closer to each other than either of them to controls (DMSO) for the SSL and the SSR models only (**Figure 5**). The distances for the ICL and WSL models were rather on the same level. Strikingly, the one-crop setup for both of them (**Figure 5**, columns 2 and 4) resulted in distances close to zero, which implies that information in the learned representations was insufficient to characterize drug effects. Multi-crop setting, in turns, caused large increase in distances, which suggests some information gain. Nonetheless, it was not enough to capture dissimilarity between drugs and controls in this case.

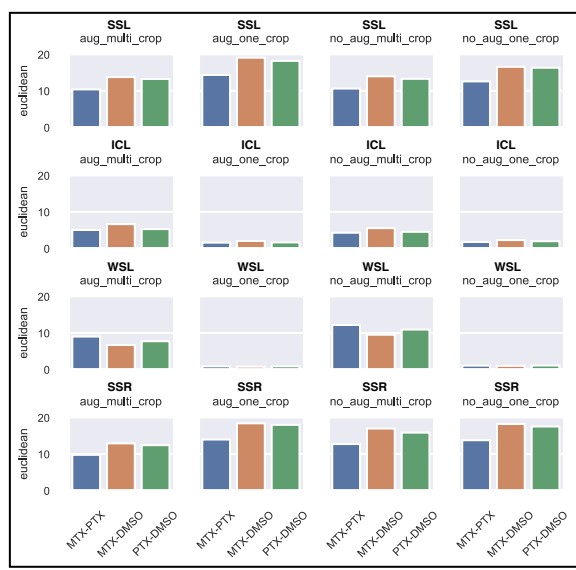

Figure 5: $D$(MTX, PTX), $D$(MTX, DMSO), $D$(PTX, DMSO) for M14 cell line.

## Appendix D. Motivation for Euclidean distance in the similarity analysis

We tested several distances to investigate how close the two drugs (PTX and MTX) were to each other and both distant from control (DMSO) in the space of learned features. We found a number of cases, where cosine and correlation distances could not differentiate between drugs and controls, i.e., $D(\text{PTX, MTX}) \approx D(\text{PTX, DMSO}) \approx D(\text{MTX, DMSO})$. Whereas Bray-Curtis and Euclidean distances both resulted in $D(\text{PTX, MTX}) < D(\text{PTX, DMSO})$ and $D(\text{PTX, MTX}) < D(\text{MTX, DMSO})$.

**Figure 6** explains it very clearly: although distributions of cosine and correlation distances are both slightly shifted towards zero

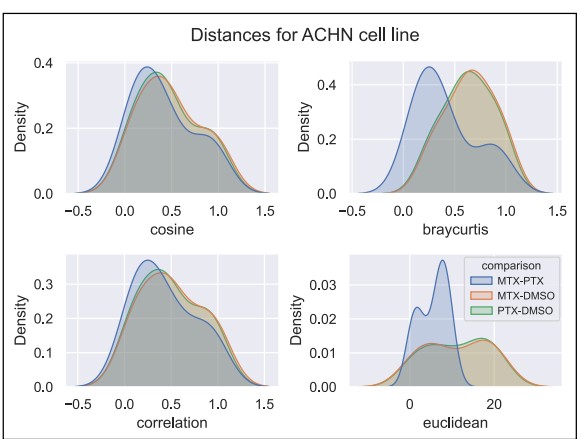

Figure 6: Distributions of distances in the similarity analysis of PTX and MTX.

for MTX-PTX comparison (blue), these effects are much stronger for Bray-Curtis and Euclidean distances. From this we concluded that Euclidean distance was the most informative for the drug similarity analysis of PTX and MTX.

## Appendix E. Discussion

In this study, we used the distance-based analysis to validate and compare models. We took images of two drugs (PTX and MTX) known to be structurally and functionally similar. We evaluated and compared their distances to control images in the space of learned features. However, this analysis stays limited to the choice of drugs. Although PTX and MTX made the best example for this dataset to use *a-priori* knowledge in validation and comparison of learned features, the results can not be generalized for any pair of drugs.

A common practice to evaluate learned representations is to apply them to different tasks and datasets. Often, linear evaluation and transfer learning scenarios are tested. However, this is the case when representations are learned from multi-class general purpose datasets (e.g., ImageNet). On the contrary, biological imaging datasets are specific. It has been reported that even SOTA models trained on ImageNet drop their performance significantly on such datasets (Grill et al., 2020). In this study, we had a large imbalanced unlabelled dataset of 1.1M cell images under 693 different conditions over time. We sampled from it in the way to formulate a balanced binary classification problem, which in turn drastically limited further transfer learning applications.

To the date, no consensual measure to evaluate clustering results has been proposed (Palacio-Niño and Berzal, 2019). A number of metrics, such as Adjusted Rand Index, Silhouette score, Normalized Mutual Information, etc., are typically used together to compare results. Most metrics, however, require the ground truth labelling, which were not available in this study. Besides, the clustering itself can be approached in many different ways, using the classical or the newly developed deep-learning based algorithms (Ciortan and Defrance,

2021). In this study, we only intended to fairly compare clustering results obtained under identical conditions (same algorithm, grid search parameters, evaluation metrics, etc.)

In this study, we have demonstrated a number of ways to analyze large biological datasets with different representation learning paradigms. Similar approaches can be applied to address actual problems in healthcare and biotech industry (e.g., deriving drug onset times, characterizing concentration-dependent pharmacodynamics, exploring opportunities for combination therapy, etc.) In this context, it is important for the scientific community to see that SOTA methods (such as BYOL) can be successfully trained on large datasets within reasonable time using limited resources.

## Appendix F. Binary classification for HT29, HCT15 and ACHN

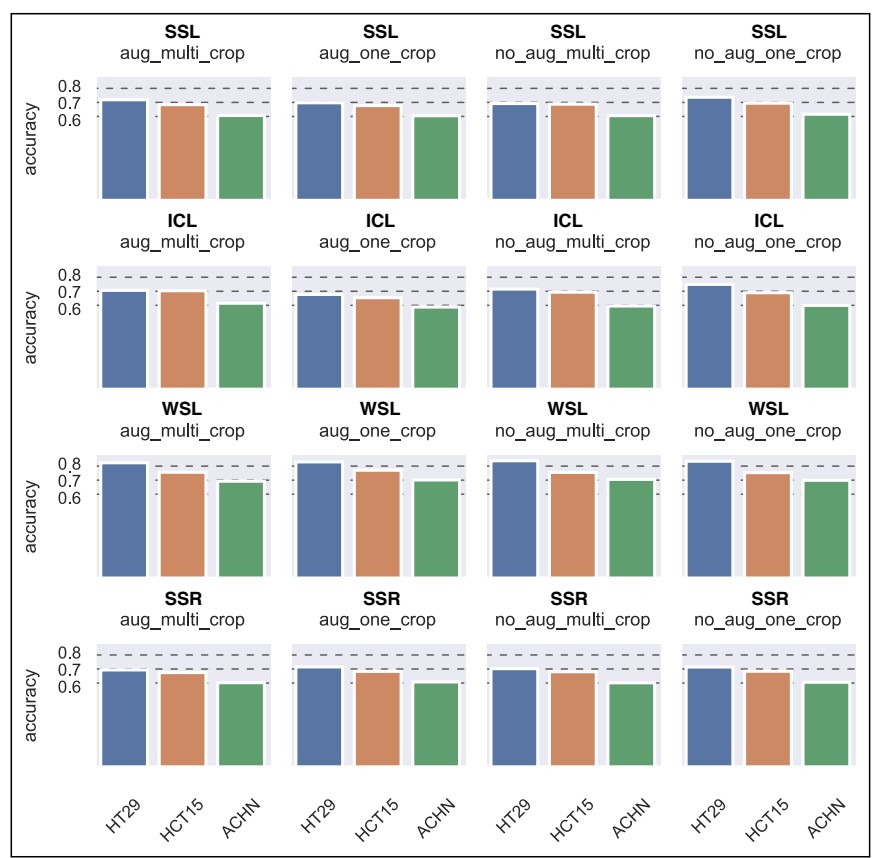

Figure 7: Binary classification accuracy (drug vs control) for three picked cell lines: HT29, HCT15, ACHN. Weakly-supervised architecture was the only one to reach 0.8 accuracy for HT29 and cross 0.7 accuracy in all settings for HCT15 and ACHN.

## Appendix G. Clustering of HCT cell line representations

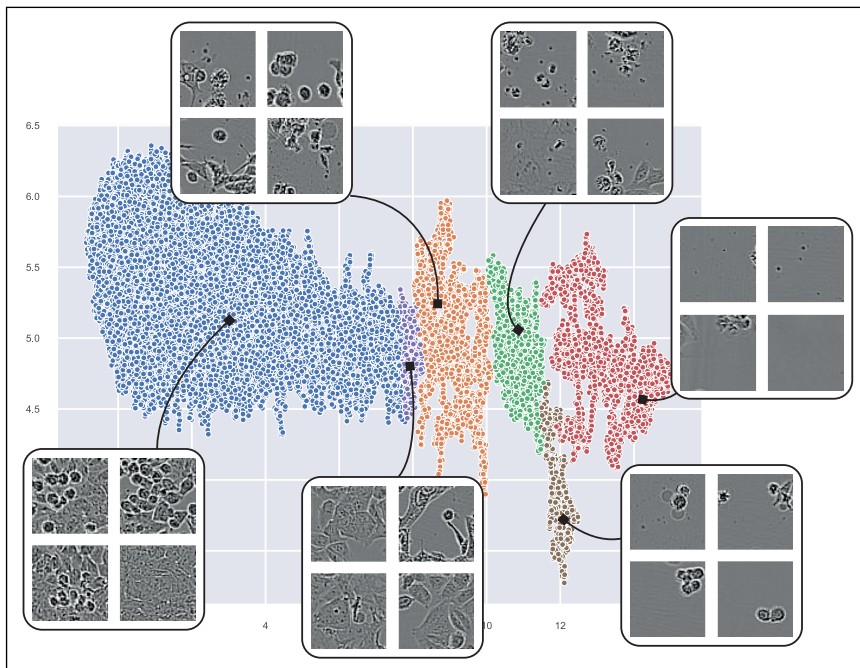

Figure 8: Clustering example of 20480 images (HCT15 cell line) with random cluster representatives. Each point is a 2D UMAP embedding of the learned image representations (the ICL model). Clusters found by HDBSCAN are highlighted in colors. The left cluster (blue) contains drugs of no effect on HCT15. The right cluster (red) contains the drugs of the strongest effect.

## Appendix H. Description of the dataset

To cover a wide range of phenotypic effects in experimental and FDA-approved anticancer drugs, we selected drugs that displayed at least 3 cell lines as resistant and 3 cell lines sensitive in the NCI-60 cancer cell line panel (**Table 3**), with a threshold in the $log_{10}$(GI50) of 1% between the sensitive and resistant groups. The list comprised 31 experimental and FDA-approved anticancer drugs, covering several modes of action of clinical and research interest (**Table 4**).

The cancer cell lines were grown in RPMI-1640 GlutaMax medium (ThermoFischer) with supplementation of 1% of Penicylin-Streptomycin (Gibco), and 5% of dialyzed fetal bovine serum (Sigma-Aldrich) at 37°C in an atmosphere of 5% $CO_2$. The seeding density to achieve a confluence of 70% was determined in Nunc 96 well plates (ThermoFischer), and that seeding density was used for experiments with a factor of four correction for the reduction in area between the 96 and 384 well plates, where cells were seeded in 45 uL of medium. Cells were incubated and imaged every two hours in the Incucyte S3 (Sartorious) 10x phase contrast mode from for up to 48 hours before drug addition, in order to achieve optimal cell adherence and starting experimental conditions. To reduce evaporation effects, the plates were sealed with Breathe-Easy sealing membrane (Diversified Biotech).

Table 3: Cell lines and inoculation densities for 96 well plate format used in the study.

| Cell line | Panel | Inoculation density |
|---|---|---|
| EKVX | Non-Small Cell Lung | 11000 |
| HOP-62 | Non-Small Cell Lung | 9000 |
| COLO 205 | Colon | 15000 |
| HCT-15 | Colon | 12000 |
| HT29 | Colon | 12000 |
| SW-620 | Colon | 24000 |
| SF-539 | CNS | 10000 |
| LOX IMVI | Melanoma | 8500 |
| MALME-3M | Melanoma | 8500 |
| M14 | Melanoma | 5000 |
| SK-MEL-2 | Melanoma | 10000 |
| UACC-257 | Melanoma | 20000 |
| IGR-OV1 | Ovarian | 10000 |
| OVCAR-4 | Ovarian | 10000 |
| OVCAR-5 | Ovarian | 15000 |
| A498 | Renal | 3200 |
| ACHN | Renal | 8200 |
| MDA-MB-231/ATCC | Breast | 20000 |
| HS 578T | Breast | 13000 |
| BT-549 | Breast | 10000 |
| T-47D | Breast | 15000 |

To allow a broad coverage of effects on time, we collected the time information about when the drugs were treated for each cell line, and corrected the analysis based on the drug treatment. Drugs were resuspended in the appropriate solvent (DMSO or water), and the same amount of DMSO (check amount) was added across all wells, including controls. The randomized 384 drug source plates were generated with Echo Liquid Handling System (Integra-Biosciences), and then transferred in 5uL of medium to Nunc 384 well plates (ThermoFischer) with the AssistPlus liquid handler (Integra Biosciences).

Table 4: Drugs, solvents, CAS registry numbers and maximum concentrations used in the study. The other four concentrations for each drug were 10x serial dilutions of the maximum concentration.

| Drug | Fluid | CAS | Concentration |
|---|---|---|---|
| Erlotinib | DMSO | 183321-74-6 | 10 $\mu$M |
| Irinotecan | DMSO | 100286-90-6 | 10 $\mu$M |
| Clofarabine | DMSO | 123318-82-1 | 10 $\mu$M |
| Fluorouracil | DMSO | 51-21-8 | 10 $\mu$M |
| Pemetrexed | Water | 150399-23-8 | 10 $\mu$M |
| Docetaxel | DMSO | 148408-66-6 | 1 $\mu$M |
| Everolimus | DMSO | 159351-69-6 | 1 $\mu$M |
| Chlormethine | DMSO | 55-86-7 | 10 $\mu$M |
| BPTES | DMSO | 314045-39-1 | 10 $\mu$M |
| Oligomycin A | DMSO | 579-13-5 | 1 $\mu$M |
| UK-5099 | DMSO | NA | 10 $\mu$M |
| Panzem (2-ME2) | DMSO | 362-07-2 | 10 $\mu$M |
| MEDICA16 | DMSO | 87272-20-6 | 10 $\mu$M |
| Gemcitabine | Water | 122111-03-9 | 1 $\mu$M |
| 17-AAG | DMSO | 75747-14-7 | 10 $\mu$M |
| Lenvatinib | DMSO | 417716-92-8 | 10 $\mu$M |
| Topotecan | DMSO | 119413-54-6 | 1 $\mu$M |
| Cladribine | DMSO | 4291-63-8 | 10 $\mu$M |
| Mercaptopurine | DMSO | 6112-76-1 | 10 $\mu$M |
| Decitabine | DMSO | 2353-33-5 | 10 $\mu$M |
| Methothexate | DMSO | 59-05-2 | 1 $\mu$M |
| Paclitaxel | DMSO | 33069-62-4 | 1 $\mu$M |
| Rapamycin | DMSO | 53123-88-9 | 0.1 $\mu$M |
| Oxaliplatin | DMSO | 61825-94-3 | 10 $\mu$M |
| Omacetaxine | DMSO | 26833-87-4 | 1 $\mu$M |
| Metformin | Water | 1115-70-4 | 10 $\mu$M |
| YC-1 | DMSO | 170632-47-0 | 10 $\mu$M |
| Etoximir | DMSO | 828934-41-4 | 10 $\mu$M |
| Oxfenicine | DMSO | 32462-30-9 | 2.5 $\mu$M |
| Trametinib | DMSO | 871700-17-3 | 1 $\mu$M |
| Asparaginase | Water | 9015-68-3 | 0.00066 units/$\mu$L |

