# OpenReview forum: "Comparing representations of biological data learned with different AI paradigms, augmenting and cropping strategies"
_MIDL.io/2022/Conference — MIDL 2022_

### Official Review · Reviewer_HrFZ · 2022-01-24

**Confidence:** 5
**Preliminary Rating:** 4
**Recommendation:** Poster

**Summary:**

This paper analyzed the effect of representation learning by controlling learning paradigms, data augmentations, and cropping strategies. The learned representation was validated in three ways: (i) class similarity, (ii) linear probe, and (iii) clustering. The downstream task was a binary classification between drug and control performed on a private dataset. The results suggested that regularized learning required more computational power but achieved the most informative representation; self-supervised learning showed comparable representation while being faster to train. Data augmentation and multiple cropping facilitated representation learning.

**Strengths:**

+ The experimental design is comprehensive and compact.

+ Many technical details can be found in the manuscript and appendix.

+ The conclusions are supported by both quantitative and qualitative results.

**Weaknesses:**

- The results were only validated on only one (private) dataset. The generalizability of the conclusions to other tasks, datasets, learning paradigms, augmentation strategies remains unclear.

- The terminology used in the manuscript is confusing and inaccurate.

**Deanonymize Review:**

no

**Detailed Comments:**

* In Figure 7, what do the numbers in the legend stand for? Why aren’t they ordered?

* The consistent terminology is very important for such an analysis paper so that the readers can easily capture the objectives and settings of the study. This terminology is better to follow the convention in the existing literature. A new word is introduced only when it is necessary (there is absolutely no other word used by peers that can describe this concept). Please improve the word choice in the paper as suggested in the answers below.

**Paper Type:**

validation/application paper

**Questions To Address In The Rebuttal:**

Overall, this paper presented an interesting analysis of four different learning paradigms and several data augmentation. In the first review round, I vote to weakly accept this work, while some inaccurate parts in the manuscript must be carefully revised.

1. The terminology in Figure 2 and Sec. 3.1 is inaccurate and confusing. In general, (a) belongs to supervised learning and (b,c,d) belong to self-supervised learning. I am not sure why the authors refer to binary classification task (a) as weakly supervised learning (see Wiki). After the invention of the concept of self-supervised learning, “unsupervised learning” (b) is no longer used for such task that is designed to learn image representation. The term “regularized learning” is confusing and not used often in the literature. Based on the description, (c) is multi-task learning and also it belongs to self-supervised learning. Finally, (d) refers to BYOL (one of the self-supervised, contrastive learning methods); it is too broad to name it self-supervised learning.

2. Terminology in the title and Sec. 3.2 is inaccurate. Multi-cropping is a type of data augmentation. The authors referred to “data augmentation” as random resized crops, horizontal ﬂips and Gaussian Blurs; referred to “multi-crop” as multiple random crops. I do not understand why cropping should be separated from data augmentation in this paper.

3. There are many terminologies that are different from what literature has been using, which caused potential miscommunication when reading the manuscript. As an example, in Sec. 3.3, the concept of “metric” is different from “downstream task” (mentioned in the abstract). To be clear, the representation was validated by three metrics (Sec. 3.3) and the downstream task was the binary classification task of drug and control. Another example, the process in Sec. 3.3.2 is not called classification metric, but “linear probing”. The linear probing has its own hyperparameter choice in the literature (e.g., number of layers, learning rate, optimizer, etc.), so the authors do not have to come up with new hyperparameters as in Sec. 3.3.2. Please follow the conventional procedure for a reproducible result.

4. In Sec. 3.2, “For multi-crop, we added 4 random resized crops applied to 64x64 images: 2 of about half-size, and 2 more of about quarter-size (5 crops in total).” What if the important image clue was cropped out? For example in Figure 1, the top right area of subfigure 1 is cut by random cropping (only background left); this may cause the model cannot tell whether it is drug or control.

5. If possible, please include at least one publicly available dataset for the reproducibility and generalizability of the conclusions drawn in the paper.

**Special Issue:**

no

---

### Official Review · Reviewer_6B68 · 2022-01-24

**Confidence:** 4
**Preliminary Rating:** 2
**Recommendation:** Poster

**Summary:**

This work presents a form of meta-study exploring four classes of existing deep learning paradigms based
on CNNs and the representations learned by these methods under different data augmentation strategies. Th
e authors study the difference in learned representations of weakly-supervised, unsupervised-, regularize
d and self-supervised learning settings. Experiments are demonstrated on 64x64 crops of grey-scale images of drug treated cancer cells. The importance of the learned representations are evaluated based on downstream classification and clustering performances.

**Strengths:**

* The overall objective of studying representation learning for image analysis under the various learning
 paradigms is interesting.
* The dataset and the downstream tasks chosen for this task are relevant and could be insightful.
* The source-code for all the models are open-source.


**Weaknesses:**

* **Comparing learned representations**: While the objective of this work is quite interesting, the experimental set-up might not be appropriate to compare the representations across different learning paradigms. By shifting the focus on comparing the representations between supervised, unsupervised and self-supervised methods with varying complexities, and in some cases differing size of latent representations, the message is somehow muddled. This work could have simply focused on studying the representations learned by each of the methods, and contrasting their peculiarities, if at all. Also, the assumption that somehow the intermediate representation in a weakly supervised method is the same as that of an unsupervised/self-supervised method is ambiguous. For instance, if the downstream task is to perform classification the learned representations at some intermediate layer is not most interesting as the final decisions could well be performed in the last layers.

* **Assumption of a Euclidean space**: The notion that the learned, high dimensional semantic space is flat (Euclidean) and distances in this space could be comparable across methods is a bit problematic [1].

* **Discussion of results**: This meta-study provides scope for discussing very interesting properties of the different methods, which in my opinion are not fully explored. By only focusing on the downstream task performance the authors overlook the impact of different learning strategies.

* Table 2 is too dense and not sufficiently well explained. Authors say that higher-is-better but in places, lower values are in bold (Ex: first row, last column is 0.20 compared to 0.17 in first column). Why is that? There are several more instances like this.
* For model b in Fig. 2, how is BCE used as reconstruction loss? The intensity values are in grey-scale, I think.

[1] Arvanitidis, Georgios, Lars Kai Hansen, and Søren Hauberg. "Latent Space Oddity: on the Curvature of Deep Generative Models." International Conference on Learning Representations. 2018.



**Deanonymize Review:**

no

**Detailed Comments:**

See points above.

**Final Rating After The Rebuttal:**

4: Weak Accept

**Justification Of The Final Rating:**

The authors have engaged with the discussions and have addressed most of the concerns I had raised in the original review. Assuming they make the revisions as committed, I am willing to raise the score to Weak Accept.



**Paper Type:**

validation/application paper

**Questions To Address In The Rebuttal:**

* Justification for comparing the representations across methods: were the size of the learned representations same? In Fig. 2-b, what is the size of learned representation?
* Can a low-dimensional visualization of clustering across methods reveal some interesting properties (like in Fig. 7 in Appendix)?
* Better explanation of results in Table 2

**Special Issue:**

no

---

### Meta-Review · Area_Chair_86dC · 2022-02-19

**Recommendation:** Accept (Poster)
**Confidence:** 3

**Metareview:**

All reviewers have brought up concerns about this study; however, after rebuttal, I think the authors have clarified a few concepts, e.g., reconstruction loss, by adding new references as well. After rebuttal, I think at least two reviewers are accepting the work and I recommend a weak acceptance to this work.

---

### Decision · Program_Chairs · 2022-02-28

Accept